# Keto Menu–Effect of Ketogenic Menu and Intermittent Fasting on the Biochemical Markers and Body Composition in a Physically Active Man—A Controlled Case Study

**DOI:** 10.3390/foods12173219

**Published:** 2023-08-26

**Authors:** Damian Dyńka, Agnieszka Paziewska, Katarzyna Kowalcze

**Affiliations:** Institute of Health Sciences, Faculty of Medical and Health Sciences, Siedlce University of Natural Sciences and Humanities, 08-110 Siedlce, Poland; damian.dynka24@gmail.com (D.D.); agnieszka.paziewska@uph.edu.pl (A.P.)

**Keywords:** ketogenic menu, ketogenic foods, ketogenic diet, intermittent fasting, caloric restriction, physical activity, body composition, low carb, high fat, ketone bodies, biochemical markers, anthropometric parameters, lipid profile, testosterone

## Abstract

The combination of ketogenic diet (KD) with intermittent fasting (IF) has, for years, aroused a great interest in the scientific world and among healthy lifestyle enthusiasts. Its importance is even greater when the study subjects are physically active individuals. The aim of the study was a determination of the effect of strict calculated ketogenic menu combined with IF and with caloric deficit on the selected biochemical markers and body composition in a 23-year-old man performing strength training. At the same time, we decided to conduct the first so-deeply investigated and controlled case study in this respect. The study protocol included a 13-week-long ketogenic diet with intermittent fasting (of delayed time-restricted eating 16:8 type) and caloric deficit. A detailed menu was designed and was used by the man throughout the whole study duration. A number of blood tests were performed before and after the implemented dietary intervention. Additionally, body composition was determined weekly and the concentrations of glucose and ketone bodies, as well as pulse rate and arterial pressure, were measured daily. The most important changes noted included a significant increase in testosterone and vitamin D concentrations and significant reduction in the HOMA-IR index and concentrations of hepatic enzymes, insulin, glucose, iron, urea, and free triiodothyronine (FT3). Moreover, a significant improvement of body composition occurred (the ratio of total body mass to the adipose and muscular tissue and water mass improved). Favourable changes were also noted in heart rate and arterial pressure values. In view of that, the KD with IF and caloric deficit exerted favourable effects on most biochemical parameters and on body composition and caused an almost twofold increase in serum testosterone concentration.

## 1. Introduction

The influence of the nutrition mode and physical activity on human body has been known since antiquity [1]. At present, together with intensification of medical examinations, increasingly frequently potential mechanisms are detected underlying the importance of diet and physical exercise for the health of every human being. The ketogenic diet (KD), although known for more than 100 years, still arouses controversies and discussions in the scientific environment. Its combination with intermittent fasting (IF) (the influence of which on health is multifaceted and increasingly frequently studied), caloric deficit, and strength sports will become an extremely interesting subject of studies in the coming years [2,3].

The ketogenic diet, in view of its character, distinguishes itself from all other diets. This results from placing the organism into nutritional ketosis state, in which ketone bodies become the main source of energy. All the remaining nutritional models are based on glucose as the main energy source [4,5]. In practice, the ketogenic diet comes down to limited consumption of carbohydrates (to 5–10% of the energy supply, usually below 50 g daily), increased share of fat in the diet (usually to 70–80% and even 90% of the energy supply), and moderate protein consumption (usually about 20% of the energy supply). Such percentages of the macrocomponents enable one to reach the state of nutritional ketosis. It consists of reorientation of the metabolism to utilising mainly ketone bodies, which are derived from fats. In view of the high-fat character of that diet, the most frequently consumed products include, inter alia, oily (and lean) fish, fatty (and lean) meat, eggs, seafood, dairy products, avocado, olives, oils (i.e., olive oil, MCT oil, linseed oil, coconut oil), nuts, nonstarchy vegetables (particularly green leafy vegetables), and, to a lesser extent, low-glycaemic fruits (among other fruits: raspberries, red whortleberries, blueberries, strawberries) [6].

Intermittent fasting is currently an extremely frequent subject of scientific research. This results from the observed multifaceted effects of the use of “nutritional window”. Certainly, all its mechanisms and detailed influence on the body are not yet known. The benefits are considered, inter alia, in the aspects of metabolic health, improvement of body composition [7,8], intestinal microbiota [9], cerebral metabolism [10], prevention of cardiovascular diseases [11], antitumour effects [12], and many other health aspects [13]. The fasting consists of temporary abstaining from eating. Various variants are distinguished, including alternate-day fasting (ADF), which means fasting every other day; twice-weekly fasting (TWF), consisting of fasting for two days weekly; early time-restricted eating (eTRE), which means eating only for 4–10 h a day (usually 8 h) in the early daytime; and delayed time-restricted eating (dTRE), i.e., as in eTRE, but the eating window is shifted to later daytime hours [7].

The aim of the study was an assessment of the effect of 13-week ketogenic diet combined with intermittent fasting (of delayed time-restricted eating type) and caloric deficit on the biochemical parameters, body composition, heart rate, and arterial pressure in a healthy, 23-year-old man performing strength training. As part of the research, we decided to conduct the first so-strictly controlled case study in this respect.

## 2. Materials and Methods

To better illustrate the methodology, a graphical diagram is provided in Figure 1.

### 2.1. Case Report

The subject of the study was a 23-year-old healthy man, not taking any drugs and performing strength training. The applied research method was an extensive, interventional, and strictly controlled case study. Table 1 shows the baseline (zero point) characteristics of the health condition.

### 2.2. Diet Applied

A two-stage menu was composed. The first stage was used for the first four weeks and the second one from week 5 to week 13. The two-stage dietary intervention resulted from the initial phase of body adaptation to using ketone bodies. Therefore, for the first four weeks, a composition of macrocomponents was used, the aim of which was to obtain the most optimal body adaptation to the state of ketosis. After four weeks, the share of carbohydrates and protein was increased and the share of fat was reduced. The caloricity at both stages of the intervention was identical (about 2450 kcal). A “nutritional window” was applied, in the form of delayed time-restricted eating (dTRE) 16:8. The first meal was taken at 13:00, the second (providing mainly fat) at 17:00, and the third one at 21:00. The interval between the last meal and sleep was 3 h on average, since the man usually fell asleep at about 24:00. The precise composition of the macrocomponents is shown in Table 2. For preparation of the menu and calculations of nutrients, Kcal-mar.pro software (accessed on 1 December 2021) was used.

The detailed amounts of the products consumed each day for the first four weeks are shown in Table 3, while Table 4 presents those for weeks 5–13.

The detailed data concerning the content of individual nutrients in the menu are shown in Table 5.

### 2.3. Physical Activity

The man, both before the study and during it, was performing strength training. The type of the workout was not changed after the beginning of the 13-week intervention (the same plan was continued as before the beginning). It included strength training (split) of medium intensity, 4–5 times weekly. The workout was usually performed in evening hours (after the second meal), and the third (the last) meal was at the same time as the post-workout meal.

### 2.4. Supplementation

The man was not taking any additional dietary supplements apart from continuation of vitamin D3 (5000 IU) and K2 (200 mcg), taken already for several months before the intervention.

### 2.5. Biochemical Parameter Measurements

#### 2.5.1. Measurements of the Biochemical Parameters before and after 13 Weeks on Ketogenic Diet

Before the application of the diet, after taking medical history and positive assessment of the health condition with absence of contraindications to nutritional intervention beginning, the man underwent blood tests at a professional Synevo diagnostic laboratory. The sample for fasting blood test was taken in the morning hours.

According to the study protocol, the set of the parameters assessed included blood count with differential, lipid profile (total cholesterol, HDL cholesterol, LDL cholesterol, triglycerides), glucose, insulin, homocysteine, alanine transaminase (ALT), aspartate transaminase (AST), gamma-glutamyl transpeptidase (GGTP), bilirubin, thyroid-stimulating hormone (TSH), free triiodothyronine (FT3), free thyroxin (FT4), creatinine, uric acid, urea, ferritin, transferrin, iron, sodium, potassium, calcium, magnesium, phosphorus, testosterone, vitamin B12, vitamin D (25(OH)D), and C-reactive protein (CRP). After termination of the 13-week intervention, all the abovementioned tests were repeated at the same laboratory (additionally, a measurement of glycated haemoglobin (HbA1c) was performed). The measurements were also performed under fasting conditions in the morning hours.

#### 2.5.2. Cyclic Measurements of the Biochemical Parameters during 13 Weeks on the Ketogenic Diet

For 13 weeks, daily after waking up, fasting measurements of ketone bodies were performed using KetoDiastrix tests (Bayer). Every day at 12:00, fasting blood glucose tests were performed.

Every third day, blood glucose level determinations were performed at 24:00 (before sleep) and blood ketone bodies levels were determined at 12:00 and 24:00 (before sleep, together with glucose level determination). The blood glucose and blood ketone bodies measurements were taken using an Optium Xido Neo FreeStyle glucometer (Abbott) with Optium Xido glucose strips (Abbott) and Optium Xido β-ketone strips (Abbott).

### 2.6. Measurements of the Anthropometric Parameters (Body Composition)

Every week for 13 weeks, under fasting conditions, detailed body composition measurements were taken. The measured anthropometric parameters included body mass (in kg), body adipose tissue content (in kg and as percentage), muscular tissue (kg and percentage), water (kg and percentage), lean body mass (kg and percentage), bone mass (kg), visceral adipose tissue (points), metabolic age (years), and body mass index (BMI). For the measurements of the anthropometric parameters, a professional TANITA SC-240 MA body composition analyser was used.

### 2.7. Measurements of the Heart Rate and Arterial Pressure

For 13 weeks, twice daily (fasting conditions in the morning and before sleep in the evening), measurements of the heart rate and systolic and diastolic arterial pressure were conducted. According to the European Society of Hypertension practice guidelines of 2021 for office and out-of-office blood pressure measurement for ambulatory blood pressure monitoring (ABPM) [15], each measurement was taken in triplicate and is expressed as the mean value. For the measurements of the heart rate and arterial pressure, an automated arm manometer (DOZ PRODUCT Basic) was used.

## 3. Results

### 3.1. Biochemical Parameters (before (P0) and after (P1) the Intervention)

#### 3.1.1. Blood Cell Count

All blood count parameters, both before and after the nutritional intervention, met the criteria of the reference ranges. The analysis of the test results demonstrated that most changes occurred in the white blood cell (WBC) values. In spite of the reduction of the total WBC count (g/L) by 15.5%, the greatest changes were observed in the percentages of individual types of blood cells. The values of the following cells increased: monocytes by 46.27%, lymphocytes by 44.22%, and eosinocytes by 41.18%, while the percentage of neutrocytes decreased by 22.68% and no change of basocyte percentage was observed. A slight change was also found in platelet count (g/L), the value of which increased by 5.82%. The changes in blood cell counts are shown in Table 6.

#### 3.1.2. The Remaining Biochemical Parameters

The biochemical parameters that underwent the greatest changes in relation to the baseline values included testosterone (significant increase by 98.3% from 24.2 nmol/L to 48.0 nmol/L), HOMA-IR index (significant decrease by 81.5% from 2.54 to 0.57), alanine transaminase (ALT) (significant decrease by 79.6% from 51 U/L to 11 U/L), insulin (significant decrease by 74.2% from 10.4 mU/L to 2.68 mU/L), vitamin D in the form of 25(OH)D metabolite (significant increase by 60.3% from 78 ng/mL to 125 ng/mL), aspartate transaminase (AST) (significant decrease by 48.1% from 27 U/L to 14 U/L), gamma-glutamyl transpeptidase (significant decrease by 42.9% from 14 IU/L to 8I U/L, iron (significant decrease by 41.6% from 202 μg/dL to 118 μg/dL), urea (significant decrease by 39.4% from 28.2 mg/dL to 17.1 mg/dL), free triiodothyronine (FT3) (significant decrease by 30.9% from 6.19 pmol/L to 4.28 pmol/L), and glucose (significant decrease by 28.3% from 99 mg/dL to 71 mg/dL).

Smaller changes concerned such biochemical parameters as total bilirubin (decrease by 22.2% from 0.9 mg/dL to 0.7 mg/dL), HDL cholesterol (increase by 18.9% from 53 mg/dL to 63 mg/dL), transferrin (decrease by 18.5% from 286 mg/dL to 233 mg/dL), free thyroxine (FT4) (increase by 17.2% from 19.22 pmol/L to 22.53 pmol/L), creatinine (decrease by 11.8% from 93 μmol/L to 82 μmol/L), and LDL cholesterol (decrease by 10.1% from 99 mg/dL to 89 mg/dL).

Slight (<10% difference in relation to the baseline values) changes occurred in such biochemical parameters as thyrotropin (TSH) (increase by 8.8% from 2.17 mU/L to 2.36 mU/L), magnesium (decrease by 5.9% from 1.7 mEq/L to 1.6 mEq/L), uric acid (decrease by 5.2% from 5.8 mg/dL to 5.5 mg/dL), potassium (increase by 5% from 4.37 mg/dL to 4.59 mg/dL), triglycerides (increase by 4.7% from 107 mg/dL to 112 mg/dL), homocysteine (increase by 3.9% from 10.2 μmol/L to 10.6 μmol/L), phosphorus (decrease by 3.2% from 3.1 mg/dL to 3 mg/dL), vitamin B12 (decrease by 3% from 540 pg/mL to 524 pg/mL), ferritin (increase by 2.2% from 181 ng/mL to 185 ng/mL), sodium (increase by 2.2% from 137 mmol/L to 140 mmol/L), total cholesterol (increase by 1.2% from 173 mg/dL to 175 mg/dL), and calcium (increase by 0.8% from 10.08 to 10.16 mg/dL).

All changes in the biochemical parameter values are shown in Table 7.

### 3.2. Biochemical Parameters Measured Periodically

#### 3.2.1. Measurements of Ketone Bodies in Blood

According to the study protocol, ketone fasting concentrations determined in blood at 12:00 every third day, already from the second measurement throughout the whole 13-week study period, were not showing values lower than 0.5 mmol/L (i.e., the value accepted as a state of ketosis, shown by the red line in Figure 1). The concentration of ketones reached the highest value on day 21 (3.4 mmol/L) and the lowest on day 71 (0.6 mmol/L). The mean of all fasting ketone values was 1.7 mmol/L.

Ketone bodies in blood measured at 24:00 before sleep every third day, already reached the concentration of 0.6 mmol/L on the first day. Throughout the whole 13-week period, the ketone values did not fall below the established limit of ketosis, i.e., 0.5 mmol/L. The highest value of ketones was measured on day 93 (3.4 mmol/L) and the lowest on days 1 and 66 (0.6 mmol/L). The mean of all ketone values measured at 24:00 was 1.8 mmol/L.

The fasting ketone values measured in blood every third day at 12:00 and before sleep at 24:00 are shown in Figure 1.

#### 3.2.2. Measurements of Ketone Bodies in Urine

The measurements of fasting ketone bodies in urine, taken daily in the morning, were presented in a previously published paper [16].

#### 3.2.3. Blood Glucose Measurements

Fasting blood glucose measurements carried out daily at 12:00 and before sleep at 24:00 were presented in a previously published paper [16].

### 3.3. Anthropometric Parameters

#### 3.3.1. Body Mass

During the 13 weeks of the nutritional intervention, the body mass decreased by 10.9 kg (from 86.8 kg to 75.9 kg), i.e., by 12.6% in relation to the baseline value. In each week the body mass was gradually decreasing (except week 11, when the value was higher by 300 g than in week 10). This is illustrated in Figure 2.

#### 3.3.2. Adipose Tissue Mass

During the 13 weeks, the adipose tissue mass decreased by 6.5 kg (from 14.8 kg to 8.3 kg), that is, by 43.9%.

In percentages, a reduction of adipose tissue amount occurred, from 17.1% to 10.9% of the total body mass.

The decreasing tendency was maintained throughout the whole study duration, although, not infrequently, the analyser was showing fluctuations of that parameter, which could have resulted from measuring imperfections of the equipment. The graphs of adipose tissue changes, both in kg and as percentages, are similar and are shown in Figure 3.

#### 3.3.3. Muscular Tissue Mass

In the 13 weeks, an increase of the muscular tissue mass occurred, from 78.8% to 84.7% of the total body mass. In spite of fluctuations of the measurement, the growing tendency was maintained. The loss of total body mass caused the percent share to increase.

The analyser showed a muscular mass reduction by 4.1 kg (from 68.4 kg to 64.3 kg), i.e., by 6%. The lowest value was achieved in week 4 (64.3 kg), below which the muscular mass never dropped. This resulted from the fact that muscle mass consists of water and glycogen, the amount of which decreases while on a ketogenic diet, particularly when combined with a caloric deficit.

The graphic representation of muscle mass changes, both in kg and percent value, is shown in Figure 4.

#### 3.3.4. Lean Tissue Mass

In 13 weeks, an increase of the lean tissue mass occurred, from 82.9% to 89% of the total body mass. In spite of measurement fluctuations, the growing tendency was maintained. The loss of total body mass and adipose tissue caused the percent share of the lean tissue to increase. The analyser showed a lean tissue mass reduction by 4.4 kg (from 72 kg to 67.6 kg), i.e., by 6.1%. This resulted from the reduction of the total body mass.

The graphic representation of lean tissue mass changes, both in kg and percent value, is shown in Figure 5.

#### 3.3.5. Water Mass

In 13 weeks, an increase of the amount of water occurred, from 57.4% to 60.9% of the total body mass. In spite of measurement fluctuations, the growing tendency was maintained. The reduction in total body mass and adipose tissue caused the percent share of the water mass in the body to increase. The analyser showed a reduction of the water mass in the body by 3.6 kg (from 49.8 kg to 46.2 kg), i.e., by 7.2%. Together with total body mass loss, the reduction in the amount of water in the body (in kg) is normal.

The graphs of changes of water amount in the body, both in kilograms and as percentages, are similar, and are shown in Figure 6.

#### 3.3.6. Bone Mass and Level of Visceral Adipose Tissue

In 13 weeks, the analyser showed a slight reduction of the total bone tissue mass by 0.3 kg, from 3.6 kg to 3.3 kg. The greatest change (−0.2 kg) was already demonstrated by the second week. The level of visceral adipose tissue decreased from level 4 to level 1.

The bone mass and visceral adipose level are shown in Figure 7.

#### 3.3.7. Body Mass Index (BMI) and Metabolic Age

In 13 weeks, a gradual decrease of the body mass index (BMI) occurred, from the baseline value 26.5 kg/m^2^ to 23.2 kg/m^2^, that is, by 3.3 kg/m^2^ (reduction by 12.5%). The analyser showed a reduction of the metabolic age from the baseline value of 20 years to 12 years.

The changes of the body mass index (BMI) and metabolic age are shown in Figure 8.

### 3.4. Arterial Pressure and Heart Rate

#### 3.4.1. Systolic and Diastolic Pressure in the Morning and Evening

During the nutritional intervention, arterial systolic and diastolic pressures both in the morning and evening hours underwent gradual changes. The values of arterial systolic pressure measured in the morning ranged from 112 mmHg to 135 mmHg (mean value 123.4 mmHg) and those measured in the evening were within the 113–136 mmHg range (mean value 123 mmHg). The values of diastolic pressure measured in the morning ranged from 59 mmHg to 84 mmHg (mean pressure 73.4 mmHg), while in the evening measurements they were from 58 mmHg to 84 mmHg (mean value 71.5 mmHg).

The mean value of arterial systolic pressure in the morning in the first four weeks was 124.8 mmHg and was higher than the mean value of that pressure from the last four weeks (122.3 mmHg). The reduction of the mean values of the diastolic pressure in the morning was even more pronounced (mean value 75.6 mmHg in the first four weeks vs. 70.9 mmHg in the last four weeks).

The mean arterial systolic pressure in the evening in the first four weeks was 125.8 mmHg and decreased to 120.9 mmHg on average in the last four weeks. The mean diastolic pressure value in the first four weeks was 75.2 mmHg and decreased to the mean value of 68.5 mmHg in the last four weeks.

The daily changes of the systolic and diastolic pressure values in the morning and evening are presented in Figure 9.

#### 3.4.2. Heart Rate in the Morning and Evening

Over the period of 13 weeks, the pulse rate both in the morning and evening underwent some gradual changes. The pulse rate values measured in the morning ranged from 64 to 95 beats per minute (BPM) (78.5 BPM on average). The pulse values measured in the evening were within the 62–92 BPM range (72.6 BPM on average).

The mean heart rate measured in the morning for the first four weeks was 80.9 BPM and decreased to 76.2 BPM in the last four weeks.

The daily changes of the pulse rate measured in the morning and evening are presented in Figure 10.

## 4. Discussion

In the light of the current literature, some of the presented results can significantly contribute to this area of scientific research. The remaining part constitutes a support for the better-grounded and better-known relationships.

### 4.1. Biochemical Parameters

#### 4.1.1. Testosterone

Among the biochemical parameters, the result that changed most was a significant increase in testosterone serum concentration. That can constitute an important contribution to this research domain, since an almost twofold increase in the concentration occurred, despite the already high values at baseline. The topic of ketogenic diet and testosterone concentration arouses controversy [17]. Reports in the literature have also demonstrated that a persisting caloric deficit can reduce testosterone levels [18]. It has been also shown that intermittent fasting alone also can reduce testosterone concentration [19]. In view of this, a supposition could seem justified that a ketogenic diet combined with intermittent fasting and caloric deficit would have a negative effect on testosterone value. However, an increase in that value was demonstrated. Particularly important is the fact that it was an almost twofold increase, from an initially high baseline value (upper limit) that, after 13 weeks, then reached the level of 172% above the laboratory limit of normal. Although many other publications also demonstrated such a relationship, it has not been on such a large scale. A study from 2023 demonstrated that after 28 days, in 22 obese men, a ketogenic diet increased testosterone concentration by 74 ± 97 ng/dL on average [20]. A meta-analysis of 2022 even suggested that a low-calorie ketogenic diet leads to a higher testosterone level increase than a normocaloric ketogenic diet [21]. A study on athletic students using a ketogenic diet for 11 weeks demonstrated a 21% increase, on average, of the total testosterone concentration (compared with students on a Western diet) [22]. This is, however, not a rule, since, e.g., Paoli et al. demonstrated a reduction in total testosterone concentration in men performing physical training and using a ketogenic diet [23]. Considering that testosterone is the main male hormone, the benefits of its sufficient concentration are well known. It performs a number of functions, including playing a key role in maintaining sexual function, affecting primary and secondary masculine characteristics. It takes part in the metabolism of proteins, carbohydrates, and fats. It also affects muscle mass, fat mass, bone mass, insulin sensitivity, lipid profile, and many others [24,25]. Wang et al. showed that high testosterone levels are inversely related to prediabetes, especially among younger men up to 50 years of age [26]. On the other hand, excess testosterone can cause side effects, e.g., in young prepubescent boys, it can lead to virilization. In adult men, side effects are particularly related to excess exogenous testosterone not naturally produced in the body [24]. In our study, the increase in testosterone was natural.

#### 4.1.2. Carbohydrate Metabolism Parameters (HOMA-IR, Glucose, Insulin, HbA1c)

Extremely significant changes were observed in the parameters of carbohydrate metabolism, i.e., HOMA-IR index and fasting glucose and insulin levels. The greatest change in this field was a significant reduction in the HOMA-IR index. In view of the fact that its value depends on glucose and insulin concentrations, the values of these two parameters also significantly improved. The results closely reflect those of the studies published thus far. Taking into account the most similar studies, it is worth mentioning the publication by Michalczyk et al. It demonstrated that in 12 weeks (thus, a similar period), the individuals using a low-calorie ketogenic diet (as in our case) decreased their HOMA-IR values from 3.73 to 1.4 on average (compared with a control group) [27]. A reduction in HOMA-IR value was also confirmed in an earlier study. It demonstrated that just four weeks on ketogenic diet are sufficient for HOMA-IR value reduction, from 2.66 to 1.44, on average [28]. In our study, a great change was observed in the case of insulin concentration, reaching the values at the lower limit of normal (2.68 mU/L, with normal range 2.60–24.90 mU/L) after 13 weeks. This may suggest a strong therapeutic effect against insulin resistance. Significant insulin concentration reductions have been observed, even after a shorter time. That was confirmed in the paper by Hernandez et al., who demonstrated that just six weeks were enough to observe a significant insulin concentration drop in individuals on a ketogenic diet (compared with the high-carb control group) [29]. Proceeding a step further, the authors of a 2018 paper demonstrated that just four days of carbohydrate restriction were sufficient to affect the glycaemia values and concentration of circulating proinsulin (Myette-Côté [30]. Many publications have confirmed a significant insulin concentration reduction after ketogenic diet application [31,32,33]. Glucose concentration reduction is, in turn, a natural reaction to strict limitation of its supply with food. The body, which can (and in some measure is forced to) use ketone bodies, makes them the main source of energy. That is why the importance of glucose is then marginal, which is reflected in a reduction in glycaemia values. Decreasing glucose concentrations during the application of a ketogenic diet have been frequently observed [30]. It has been shown that a restriction of carbohydrates alone was able, even over a long time period, to significantly reduce HOMA-IR value and fasting insulin concentration by 43% on average, and 73% of that insulin drop was observed in the first 70 days [34]. The correct glycaemia values during 13 weeks of our study are best demonstrated by the glycated haemoglobin level determination performed after that period. The demonstrated Hb value of 4.7% (with normal range of <6% Hb) confirms the mean glucose concentration during that period at the level of half of the fasting reference range. The significant effect of a ketogenic diet on glycaemia and insulin concentration values is best reflected by the fact that the diet is increasingly frequently suggested as a possible and effective therapeutic solution for diabetic patients [35]. An effect on the reduction in glycaemia and insulin concentration was also certainly exerted by caloric deficit and possibly by intermittent fasting. This is supported by recent publications suggesting an effect of intermittent fasting on glycaemia control and improvement of sensitivity to insulin [36,37]. In view of this, it can be assumed that a ketogenic diet combined with intermittent fasting and caloric deficit can more favourably affect glycaemia control than only one of the factors mentioned [38].

#### 4.1.3. Liver Function Parameters (ALT, AST, GGTP, Bilirubin)

The results of studies negate the common erroneous opinion that a ketogenic diet contributes to liver function impairment. The significant reduction in the values of these parameters in our study can, rather, be evidence of a significant relieving of the workload of the organ. The results are important in the context of the currently ongoing scientific debates on the effect of ketogenic diet on liver function. This is due to the fact that the data are not unequivocal. A 12-week study showed that a Mediterranean ketogenic diet contributed to average reduction of alanine transaminase (ALT) level from 71.9 U/L to 37 U/L and of aspartate transaminase (AST) concentration from 47.71 U/L to 29.57 U/L. The study was conducted in obese males with nonalcoholic fatty liver disease (NAFLD) and multiple sclerosis [39]. A beneficial effect on the liver function parameters, such as ALT or AST, was also demonstrated in a study of 2020, based on the example of a low-calorie ketogenic diet with whey protein version. An ALT and AST level reduction occurred after only 45 days [40]. Although more studies are available, confirming this relationship [41], some do not show it. In the study by Paoli et al., no significant differences were demonstrated in ALT, AST, and GGTP concentrations after six weeks on Mediterranean ketogenic diet [42]. No clinically significant changes of liver function were also shown in a study of 2020 [43]. The bilirubin concentration reduction observed in our study is also an interesting finding, since its value dropped below the lower limit of normal. It is known from the literature that even in such conditions as Gilbert’s syndrome, a ketogenic diet is able to decrease bilirubin concentration [44]. A wider context of ketogenic diet and liver function was described in the publication by Watanabe et al. [45]. This may also be influenced by intermittent fasting, as the publication by Badran et al. showed that intermittent fasting reduced ALT, AST, and GGTP [37].

#### 4.1.4. Iron Metabolism Parameters (Iron, Ferritin, Transferrin)

Of interest is also the change of iron level in blood. In spite of the presence of meat and eggs in the diet, this value decreased significantly. This is, however, not unfavourable in this case, since before diet application, the concentration of iron exceeded the reference values. After 13 weeks on the ketogenic diet, the blood level of that element reached the normal value. The amount of protein transporting iron in the body, i.e., transferrin, also decreased. That seems obvious at the time of reduction in the level of iron itself. Furthermore, the concentration of ferritin, the protein responsible for iron storage in the body, was unchanged and even slightly increased. That may suggest a significantly better efficiency of the iron metabolism of the body. Similar changes were noted in the study by Klement et al., in which 5–7 weeks of application of ketogenic diet in sports-practicing individuals led to an iron concentration reduction from 90 µg/dL to 72 µg/dL on average. A simultaneous increase of ferritin level was shown, from 54 ng/mL to 88 ng/mL on average [46]. Some difference between ketogenic and high-carb diets was also suggested by a study of 2019 conducted in athletes. It was demonstrated that, in spite of lower iron content in ketogenic diet (by 25% on average), the ferritin level reduction was lower (23%) compared with a high-carb diet, containing more iron (reduction by 37%) [47]. On the other hand, the study by Kose et al. showed no significant changes in the concentrations of iron, ferritin, and transferrin in individuals using a ketogenic diet for 12 weeks [48]. The potential changes of iron concentration may also depend on caloric deficit or physical activity [49]. This is described in a wider context in a publication of 2020 [50].

#### 4.1.5. Thyroid Function Parameters (TSH, FT3, FT4)

Contrary to the majority of the remaining results, 13 weeks on a ketogenic diet with intermittent fasting and caloric deficit failed to demonstrate any benefit in the functioning of the thyroid. This can result not from the ketogenic diet itself, but from the caloric deficit and “nutritional widow”, which was shifted to the later part of the day. Although it is not a favourable effect, our finding is confirmed in the literature. Comparing our results with others, described in the literature, it is worth mentioning the study by Iacovides et al. It demonstrated that in healthy individuals, a ketogenic diet significantly reduced triiodothyronine values and increased thyroxin levels compared with those on a high-carb diet. Importantly, the authors noted no significant changes in thyrotropin concentration, which, in our study, also did not change by more than 10% [51]. Other, similar results were reported in the study by Multberg et al. It was conducted for 12 weeks in adult epileptic patients using ketogenic diet. A reduction of FT3 value by 10.6%, T3 concentration by 13.4%, an increase of FT4 value by 12.1%, and a slight increase in TSH concentration were noted [52]. The effect of intermittent fasting alone on thyroid hormones has also been observed in the studies published. One of them revealed that during eight weeks, in trained and fit men, the intermittent 16:8 fasting led to a triiodothyronine level reduction of 10.7% [53]. Moreover, caloric deficit also plays an important role in potential changes of thyroid hormone levels [54,55]). Therefore, it is no surprise that a combination of all these three elements together is able to change the picture of the mentioned parameters.

#### 4.1.6. Lipid Profile (Total Cholesterol, HDL, LDL, Triglycerides)

Although frequently, in view of its high-fat character, a ketogenic diet has been associated with a worsening of blood lipid profile, multiple benefits resulting from its application were demonstrated as part of the research project. This is evidenced by the absence of significant changes in total cholesterol and triglyceride concentrations, and by the increase in HDL lipoprotein and the decrease in LDL lipoprotein levels, still within the official normal ranges. Improvements were demonstrated of the HDL cholesterol to LDL cholesterol ratio from 0.54 to 0.70, total cholesterol to HDL ratio from 3.26 to 2.78, and triglyceride to HDL cholesterol ratio from 2.02 to 1.78. That example is important evidence that it is not the total supply of fat or even cholesterol (not infrequently >1000 mg daily) that exerts a significant effect on the lipid profile, but other factors are responsible. This occurred also despite the supply of saturated fat in amounts significantly exceeding the standard dietary recommendations (<10% of the total energy share). Taking, however, into account the literature, the results differ significantly, and on their basis, no unequivocal conclusions can be drawn. In a 56-week study in individuals using a ketogenic diet, reductions in total cholesterol, triglyceride, and LDL concentrations were noted, with a simultaneous increase in HDL lipoprotein concentration [56]. More and more frequently, potential benefits related to the diet are reported, such as improvement of the lipid profile, i.e., reduction in LDL and triglyceride concentrations and increase in HDL level [57]. Negative results are frequently observed in shorter-lasting studies; therefore, it is suggested that a ketogenic diet can only initially impair the lipid profile, and after some time it normalises these parameters. A study of 2021 demonstrated that in four weeks an increase in LDL fraction level occurred in healthy young women using a ketogenic diet [58]. Another short 5–7-week observation revealed an increase in total cholesterol concentration (from 204 mg/dL to 277 mg/dL on average), LDL concentration (from 116 mg/dL to 157 mg/dL on average), and a slight increase in triglyceride level (from 64 mg/dL to 76 mg/dL) in sports-training individuals. In spite of that, even in that study, an increase in HDL fraction was demonstrated (from 92 mg/dL to 104 mg/dL on average), which suggested that not only unfavourable consequences occurred [46]. These types of short-term observations frequently demonstrate some negative effects of the ketogenic diet. Another confirmation of this fact is the six-week-long study in which a ketogenic diet applied to healthy adults slightly increased the total cholesterol (+4.7%) and LDL fraction (+10.7%) concentrations. HDL fraction and triglyceride values remained unchanged [59]. This shows that the results of studies are equivocal, which may result from the mode of study conduction, ketogenic diet applied, and, mainly, from the duration of such studies. This may also be influenced by intermittent fasting, as the publication by Badran et al. showed that intermittent fasting lowered total cholesterol, triglycerides, and LDL, and increased HDL [37].

#### 4.1.7. Renal Function Parameters (Urea, Creatinine, Uric Acid)

The observed biochemical parameters related to the functioning of the kidneys suggest a favourable effect of a ketogenic diet on the functioning of that organ. A significant reduction in urea level may be of particular importance from the perspective of scarce amounts of data in the literature on that topic. A urea level reduction due to ketogenic diet application was also observed in a 56-week-long study [60]. On the other hand, in a 24-week-long study, no significant changes were demonstrated in the concentrations of both urea and creatinine [61]. A meta-analysis of 2022 demonstrated, however, no significant effect of a ketogenic diet on urea and uric acid concentrations in blood, although the publication concerned patients with tumours [62]. The effect of a ketogenic diet on uric acid level has not been fully elucidated. The paper by Baloch et al. even suggests using a ketogenic diet for gout [63]. A potential mechanism is suggested that could be responsible for uric acid concentration reduction in gout; this is associated with the effect of β-hydroxybutyrate, which inhibits NLPR3 inflammasome [64]. On the other hand, the effect of the diet on creatinine concentration has also not been studied in detail. In our study, it was slightly reduced; in another, 24-week-long study, no significant changes were observed [65], and in yet another one, an insignificant increase was demonstrated [46]. Taking into account the not-unequivocal study results, our findings make an important contribution to this research area.

#### 4.1.8. Vitamin D 25(OH)D

Among the remaining parameters, an unusually significant concentration change was observed in the case of vitamin D in the form of 25(OH)D in serum. This could have been a result, on the one hand, of the supplementation administered (although it was merely a continuation of the preinterventional supplementation) and, on the other, of body mass reduction. In addition to that, the high-fat character of the diet ensures a better absorbability of vitamin D. That has been confirmed in the available literature. In the study by Perticone et al., body mass reduction due to ketogenic diet resulted in a vitamin D concentration increase from 18.4 ng/mL to 29.3 ng/mL on average, i.e., by 59.2%, over 12 months [66]. This suggests a relationship between concentration of the vitamin and body weight, BMI, waist circumference, and adipose tissue level. In a study from 2021, a close relationship was demonstrated between reduction in the body mass and adipose tissue mass alone and increase in vitamin D concentration. With body mass reduction by 15.1 kg, vitamin D concentration grew by 4.2 ng/mL on average [67]. In diabetic patients on a ketogenic diet, the concentration of vitamin D was significantly higher (mean concentration 53.5 ng/mL) than in those not using a ketogenic diet (mean concentration 25.1 ng/mL) [68].

#### 4.1.9. Microelements (Sodium, Potassium, Calcium, Magnesium, Phosphorus)

The concentrations of sodium, potassium, calcium, magnesium, and phosphorus were not changed significantly. This possibly resulted, on the one hand, from the optimally designed and calculated menu and, on the other, from the sufficiently long study duration (13 weeks). In view of the initial period of ketoadaptation, the electrolyte concentrations can be significantly lowered, but they return to normal after a short time [6]. In view of the paramount importance of the method of composing the ketogenic menu and amounts of microelements, a comparison of the results with those of other authors is difficult. Trying, however, to make a comparison with the available publications, it is worth mentioning, among other papers, the study by Kenig et al., in which some interesting results were observed. In spite of consumption of lower (in relation to the recommendations) amounts of potassium, magnesium, calcium, phosphorus, and iron, after 12 weeks, the concentrations of all of them were maintained within the reference range (with the exception of calcium, the concentration of which dropped by 6.3%) [69]. No changes in the concentrations of sodium, calcium, and potassium (and even a slight increase) were also reported in another paper concerning a 5–7-week study in healthy, sports-practicing individuals [46]. The concentration of magnesium may slightly decrease [70], although maintaining the concentration of this element in the norm is not a problem. In addition to that, an inverse relationship has been observed between magnesium concentration and glucose level [71].

### 4.2. Body Composition

Because of the nutritional intervention applied, evident changes in body composition were observed. Lower body mass, body mass index (BMI), adipose tissue amount, and greater percent share of lean body mass, of muscle mass, and of water in the body are evidence of a significant improvement of the body composition. Taking into account simultaneous combination of ketogenic diet with caloric deficit, strength training, and intermittent fasting, such effects seem understandable. Indeed, the deficit alone will change the body mass and composition. In addition to that, taking into account the mode of composition of the ketogenic diet, its caloric value, type of physical activity, characteristics of the studied individuals, and study duration, it is difficult to unequivocally compare results of one study with those obtained by other authors. However, based on the available studies, which show a potential direction of changes, it is worth referring to several publications. Citing similar study reports, among other papers, a publication of 2022 can be mentioned. It assessed the effect of a very-low-calorie ketogenic diet with omega-3 acid supplementation, among other parameters, on body composition in obese individuals. After 90 days, the study participants decreased their body mass by 13.7 kg, on average, of which the mass of the adipose tissue alone was 9.1 kg on average. Furthermore, there were an evident reduction of the waist circumference by 13.3 cm and of visceral adipose tissue mass by 0.41 kg [72]. Even after a significantly shorter time (5–7 weeks), a ketogenic diet with caloric deficit in physically active individuals reduced the adipose tissue mass from 16.4 kg to 13 kg on average and an increase of the lean body mass was observed, from 57.4 kg to 58.6 kg on average. The total body mass, with body composition change, decreased from 73.7 kg to 71.4 kg on average [46]. Comparing the effect of a ketogenic diet (KD) with a Western diet (WD) in natural bodybuilders during two months, some interesting results were also found. The total body mass slightly decreased in the KD group (86.39 ± 15.42 kg to 85.51 ± 13.62 kg) and slightly increased in the WD group (89.04 ± 11.73 kg to 90.37 ± 9.91 kg). The mass of adipose tissue alone decreased in the KD group from 9.86 ± 3.79 kg to 8.42 ± 2.41 kg, and in the WD group from 10.60 ± 3.92 kg to 9.70 ± 2.53 kg. The lean body mass increased in the KD group from 76.53 ± 12.13 kg to 77.09 ± 11.47 kg, and in the WD group from 78.44 ± 8.31 kg to 80.67 ± 7.72 kg. Some advantage of the ketogenic diet was thus demonstrated, with respect to body composition change. It also is worth mentioning here that the total caloric value in both groups was similar (KD: 3443.70 ± 545.94, WD: 3529.71 ± 374.06), as was the amount of protein (KD: 215.97 ± 38.55 g; WD: 222.60 ± 29.33 g). That suggests a high reliability of the results obtained [23]. Reliable results can be also found in the study by Yancy et al., in which the authors compared the effects of a ketogenic diet with those of a low-fat diet, on, inter alia, body composition over a period of 24 weeks. Importantly, in both groups, the participants consumed similar amounts of calories (1461.0 ± 325.7 kcal on the ketogenic diet and 1502.0 ± 162.1 kcal on the low-fat diet). Body mass loss was almost twice higher in the participants on the ketogenic diet and it decreased by 12.9% on average, while in the individuals on the low-fat diet, the body mass reduction accounted for 6.7% on average. An almost twofold difference also occurred concerning the adipose tissue, since its reduction was demonstrated by 9.4 kg on average in the ketogenic group, compared with 4.8 kg in the low-fat group. The lean tissue mass decreased by 3.3 kg in the individuals on the ketogenic diet, compared to 2.4 kg in the other group, and in the group observing the ketogenic diet rules, a significant increase in HDL cholesterol concentration was also observed [73]. Studies are also available comparing a ketogenic diet with a standard diet, while the caloric value of the ketogenic diet is significantly lower. Not surprising, therefore, is the significantly greater body mass loss when fewer calories are consumed. In a paper from 2020, the effects were compared between a very-low-calorie ketogenic diet (600–800 kcal daily) and a standard balanced low-calorie diet (1400–1800 kcal daily). After two months, a significant advantage was observed for the ketogenic diet, which caused body mass reduction by 9.59 kg on average, compared with the other group, in which only 1.87 kg was lost. The ketogenic diet also contributed to a significant reduction in the amount of liver fat [74]. Therefore, it is also worth placing the results of all studies into a wider context. A publication of 2021 summarised the analysed results on that subject. The authors suggested that when ketogenic diets are compared with standard ones, and if both types include a similar deficit, then no significant differences are observed with respect to their effect on body composition [75]. It is, however, difficult to establish the influence of the “nutritional window” alone on the effects obtained. Some publications suggest that it offers more advantages than meals consumed in the standard way with respect to body composition improvement [76], while other papers show no major differences. This has been described, inter alia, in the paper from 2023, informing that no differences were found in adipose tissue mass or lean body mass between the interventions in already trained and fit young men [77].

### 4.3. Arterial Pressure and Pulse Rate

In view of few studies on the effect of ketogenic diet on arterial pressure and pulse rate (particularly strictly and regularly monitored), our results constitute a significant contribution to better understanding of the influence of a ketogenic diet on these parameters. This is of even greater importance, since the results suggest a gradual reduction in the pulse rate and arterial pressure, both systolic and diastolic, in the morning and evening hours, due to a ketogenic diet combined with intermittent fasting and caloric deficit. A need for further studies was explicitly suggested also by the authors of a paper from 2021, including an analysis of the influence of a ketogenic diet on arterial pressure values [78]. Based on the available data, the authors formulated the thesis that a ketogenic diet is able to reduce arterial pressure values, but not to a significantly greater extent than other nutritional models. This suggests, thus, a hypotensive effect mainly of body mass reduction and not of the ketogenic diet itself. However, based on the available studies, which concern the effect of carbohydrate consumption reduction on arterial pressure, the data rather suggest favourable effects of a ketogenic diet on arterial pressure values. An extraordinarily extensive study on that topic is the publication by Foster et al. The authors studied the long-term (3, 6, 12, and 24 months) effect of a low-carb diet on arterial pressure compared with a low-fat diet. In the low-carb group, in the first three months, the systolic and diastolic arterial pressure decreased by 7.74 mmHg (systolic) and 5.53 mmHg (diastolic) on average, compared with 5.20 mmHg (systolic) and 3.05 mmHg (diastolic) in the low-fat group. After six months, the pressure values in the individuals on low-carb diet decreased by 7.36 mmHg (systolic) and by 5.15 mmHg (diastolic), compared with 6.97 mmHg (systolic) and 2.5 mmHg (diastolic) in the low-fat diet group. After 12 months, the systolic pressure decreased by 5.64 mmHg and diastolic pressure by 3.25 mm in the low-carb group, while in the individuals on a low-fat diet, the systolic pressure was reduced by 4.06 mmHg and diastolic pressure by 2.19 mmHg. After two years (24 months) in the low-carb group, the mean reduction of systolic pressure was 2.68 mmHg, and that of diastolic pressure was 3.19 mmHg. In the low-fat group, the reductions reached 2.59 mmHg (systolic) and 0.5 mmHg (diastolic). Moreover, the authors observed an increase in HDL cholesterol concentration in the individuals of low-carb diet [79]. A paper from 2020 also demonstrated an effect of various types of ketogenic diet (with whey protein vs. plant protein vs. animal protein), effectively reducing arterial pressure after just 45 days. Irrespective of the version used, in each one, a blood pressure reduction occurred. It included a reduction from 132/78 mmHg to 124/70 mmHg on average in the group with whey protein, from 131/78 mmHg to 121/72 mmHg in the plant protein group, and from 129/78 mmHg to 121/71 mmHg in the animal protein group [40]. Taking these results into account, the effect of a ketogenic diet on blood pressure is visible, although, currently, the number of studies is insufficient to prove that unequivocally. Furthermore, in spite of the absence of an unequivocal answer to the question of the effect of intermittent fasting alone on the parameters mentioned, some role of it in the observed results of pulse rate and arterial pressure values cannot be ruled out [80,81].

## 5. Summary

The obtained results are an important element in the understanding of the multifaceted effects of a ketogenic diet on the human body. A number of changes in the biochemical and anthropometric parameters, pulse rate, and blood pressure are closely interrelated. On the one hand, the state of ketosis itself affects many blood parameters, e.g., glucose and insulin concentrations, HOMA-IR index, lipid profile, etc. On the other hand, the caloric-deficit-induced change of body mass and composition also exerts influence on many biochemical parameters and pulse rate and blood pressure values. The application of intermittent fasting, the effect of which on the body is multifaceted, certainly contributed to the final results. Strength training undoubtedly influenced the specificity of body composition changes and this, in turn, is important for the picture of the remaining parameters. An integrated combination of such elements as ketogenic diet, caloric deficit, intermittent fasting, and physical activity contributed to the finally obtained favourable effects, health condition, and the presented results of tests in the man. The greatest changes were observed in the concentration of testosterone, HOMA-IR index, alanine transaminase (ALT), insulin, vitamin D, aspartate transaminase (AST), gamma-glutamyl transpeptidase (GGTP), urea, free triiodothyronine (FT3), and glucose. The increase in testosterone concentration could have resulted from strength training, body mass loss, and from the high-fat balanced menu. Similar was the HOMA-IR index, which reflects serum glucose and insulin concentrations. A low-carb diet causes a decrease in glucose concentration, which leads to insulin concentration reduction. This is of particular importance if it is correlated with caloric deficit. Strength training activates the utilisation of glycogen reserves, stimulates muscular mass increase, and causes loss of adipose tissue. The expenditure of a greater amount of energy than is supplied leads to body mass loss and change of body composition. This, in turn, also affects the glycaemia values. The fact is that reduction in hepatic enzyme concentrations is also correlated with body mass loss and, as demonstrated, with a reduction in visceral adipose tissue. Correct composition of the diet, physical activity, and body mass reduction exert effects on all other observed parameters, e.g., related to renal function or on lipid profile. Certainly, it can thus be said that all the obtained results were affected by each of these elements separately, as well as by the integral entirety of their complex combination. For this reason, their summed-up effect on the course of changes of all parameters is most likely different than could be suggested only by summing up of their effects. Potential interactions between a ketogenic diet and intermittent fasting are illustrated on Figure 11.

The described multifaceted analysis is the first published case study of its type, in which not only is a specific dietary intervention proposed and presented, but a controlled clinical observation is also conducted with consideration of biochemical parameters, body composition, pulse rate, and arterial pressure.

In conclusion, the 13-week-long nutritional intervention with the use of a ketogenic diet combined with intermittent fasting and caloric deficit exerted a favourable effect on the general health condition, expressed by the presented biochemical parameter values, body composition, pulse rate, and arterial pressure values in the studied man.

## Data Availability

The data used to support the findings of this study can be made available by the corresponding author upon request.

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
