# Peer review of "Keto Menu–Effect of Ketogenic Menu and Intermittent Fasting on the Biochemical Markers and Body Composition in a Physically Active Man—A Controlled Case Study"

_foods, 2023, doi:10.3390/foods12173219_

Round 1

Reviewer 1 Report

The article by Kowalcze et al. is well-planned, and its presentation is logical, reader-friendly, and graphically sound. To keep praising the authors is excessive and uninformative. Upon publication (which I recommend as a reviewer), I will appreciatively refer to the paper’s data as a co-author in an upcoming summarising review of designs, effects, and mechanisms of ketogenic diets. 

Thomas Seyfried remarked in his book (1., page 6): “The definition of ketogenic diet allows for considerable leeway in food choices as long as the individual has reduced blood glucose and is producing ketones.”  Accordingly,  there are two major approaches to researching and implementing ketogenic diet(s): design and/or analysis of a) diet composition and b) diet effects. The paper takes advantage of both, which is a regretfully rare case.

  1. Seyfried TN. Cancer as a Metabolic Disease: On the Origin, Management, and Prevention of Cancer. Hoboken, NJ: Wiley;  (2012)

Reviewer 2 Report

Dynka et al. present an interesting study on the role of the combination of ketogenic diet and intermittent fasting (KD/IF) on biochemistry and body composition in a young healthy male. The main drawback of the study is too small group (one participant) in order to make any conclusions. The study is well-designed, and the results are promising, but there are a few areas where the authors could improve the manuscript.

The authors should justify the selection of the participant. They selected a young healthy male, but it would be interesting to see the effect of KD/IF on different age groups and sex, and also KD and IF separately.

Some of the blood parameters are on the borderline (LDL, glucose, total bilirubin, hemoglobin) or above the reference range (ALT, iron, calcium). The authors should investigate whether these changes are within the normal range for this participant or if they could be indicative of a health disorder.

The discussion section is too long and could be shortened. The authors should focus on their own findings and discuss the potential mechanisms by which KD/IF could be influencing the metabolism and physiology of the participant.

The main finding of the study is the significant increase of testosterone levels above the reference range. This is an interesting finding, but it is not clear what the long-term implications of this are. The authors should discuss the potential risks and benefits of elevated testosterone levels, especially in young men.

Finally, the authors could provide a figure with potential mechanisms of KD/IF influence on the metabolism/physiology. This would help to visualize the complex interactions between these two interventions and the body's systems.

Minor editing

Reviewer 3 Report

The study presents an intriguing exploration of the combined effects of a ketogenic diet (KD) and intermittent fasting (IF) on a physically active individual's biochemical markers and body composition. The focus on a 23-year-old man engaged in strength training adds relevance to the investigation. By employing a meticulously calculated ketogenic menu coupled with IF and caloric deficit, the study aims to shed light on the potential impacts of this combination. Notably, the researchers have undertaken a comprehensive approach, emphasizing a controlled case study of significant depth.

Why did the study opt for a single-subject case study design instead of a larger sample size?

Why was a control group not included to compare the effects of the ketogenic diet with intermittent fasting and caloric deficit against alternative dietary approaches?

Why was a 13-week duration chosen for the study, and how might this duration impact the interpretation of results?

Why were the specific biochemical markers (testosterone, vitamin D, hepatic enzymes, etc.) selected for analysis, and how do these markers collectively contribute to the study's objectives?

Why is it important to highlight the novelty and depth of investigation as a key strength of the study, and how might this contribute to the broader scientific understanding of the combined effects of KD and IF on biochemical markers and body composition?

The quality of English language in the provided text is generally good
